# GeniL: A Multilingual Dataset on Generalizing Language

**Aida Davani,**[*] **Sagar Gubbi Venkatesh,**[*] **Sunipa Dev, Shachi Dave,**[†]
**Vinodkumar Prabhakaran**[†]
Google Research
{aidamd,gubbi,sunipadev,shachi,vinodkpg}@google.com

## Abstract

Generative language models are increasingly transforming our digital ecosystem, but they often inherit societal biases learned from their training data, for instance stereotypes associating certain attributes with specific identity groups. While whether and how these biases are mitigated may depend on the specific use cases, being able to effectively detect instances of stereotype perpetuation is a crucial first step. Current methods to assess presence of stereotypes in generated language rely on simple template or co-occurrence based measures, without accounting for the variety of sentential contexts they manifest in. We argue that the sentential context is crucial to determine if the co-occurrence of an identity term and an attribute is an instance of generalization. We distinguish two types of generalizations —(1) where the language merely mentions the presence of a generalization (e.g., "people think the *French* are very *rude*"), and (2) where the language reinforces such a generalization (e.g., "as *French* they must be *rude*"—, from a non-generalizing context (e.g., "My *French* friends think I am *rude*"). For meaningful stereotype evaluations, we need scalable ways to reliably detect and distinguish such instances of generalizations. To address this gap, we introduce the new task of detecting generalization in language, and build GeniL, a multilingual dataset of over 50K sentences from 9 languages —English, Arabic, Bengali, Spanish, French, Hindi, Indonesian, Malay, and Portuguese— annotated for instances of generalizations and their types. We demonstrate that the likelihood of a co-occurrence being an instance of generalization is usually low, and varies across different languages, identity groups, and attributes, underscoring the inadequacy of simplistic co-occurrence based approaches. We also build classifiers that can detect generalization in language with an overall PR-AUC of 58.7, with varying degrees of performance across languages. Our research provides data and tools to enable a nuanced understanding of stereotype perpetuation, a crucial step towards more inclusive and responsible language technologies.

## 1 Introduction

Stereotyping is a cognitive bias with a pervasive presence in human language (Beukeboom and Burgers, 2017; Charlesworth and Banaji, 2022). Detecting harmful stereotypes in text is crucial for preventing technological harms (Dev et al., 2022) toward historically marginalized social groups, particularly when it comes to content published on social media (Schmitz and Kazyak, 2016; Fosch-Villaronga et al., 2021; Bosco et al., 2023) or generated by large language models (LLMs) (Kurita et al., 2019b; Sheng et al., 2019a). Natural Language Processing (NLP) benchmarks of stereotypes are commonly designed to cover real-world *stereotypical associations* in the form of identity group and attribute pairs, e.g., "*Man*" and "*Computer Programmer*" (Caliskan et al., 2017; Rudinger et al., 2018; Zhao et al., 2018a; Jha et al., 2023). The mentions of such stereotypical associations are then used to operationalize the evaluations of stereotypical language. However, such operationalizations of stereotypes, although rich

---

[*]equal contribution
[†]equal contribution

| | Sentence | Association | GeniL Label |
|---|---|---|---|
| EN | Zigun's fiancee, Pat Mukuro, is not only a Nigerian pop singer, but, if you believe the entirely untrustworthy Zigun, she's also an Urhobo princess. | Nigerian, untrustworthy | NG |
| EN | There has been a lot of anti-Chinese sentiment, and often for similar reasons: "they are economically successful, they are pushy, they are clannish." | Chinese, clannish | MG |
| EN | Bargaining and staring aside, Nepalese people are the friendliest, most helpful and lovely people. | Nepalese, helpful | PG |
| AR | ومن بين أسباب انطفاء شعلة العشق الإفريقي لفرنسا، الدعم الصريح الذي تقدمه باريس لبعض القادة المستبدين حيث تواجد القواعد العسكرية الفرنسية يقوي الشعور لدى المواطن الإفريقي بأن فرنسا لا تتحرك سوى لحماية الحكومات المتسلطة تحت ذرائع واهية | الفرنسية, المتسلطة | MG |
| BN | রাশেদ খান মেনন ধর্মপ্রাণ মুসলমানদের ঈমানী আন্দোলনকে পাকিস্তানি যোগসাজশ দেখিয়ে মূলত অমুসলিম কাদিয়ানিদের পক্ষ নিয়েছেন | পাকিস্তানি, ধর্মপ্রাণ | MG |
| ES | Los rusos beben mucho vodka, no es un mito | rusos, vodka | PG |
| FR | Aux Seychelles, archipel situé dans le sud-ouest de l'océan Indien, la pêche constitue la principale source de nourriture et de revenu pour de nombreuses petites communautés comme à La Retraite sur l'île de Mahé | Seychelles, archipel | MG |
| HI | कई वर्षों से, मीडिया ने पद्धति और दृढ़ता के साथ हमें हमारे सिर में डाल दिया है कि रूस एक शाश्वत हारे हुए हैं, कि हमारे पास केवल "मूर्ख और सड़कें" हैं, रूसी शराबी और आलसी हैं और अन्य "सज्जनों के रसोफोब्स सेट हैं | रूसी, शराबी | MG |
| ID | Budaya oleh-oleh sudah jadi budaya orang Indonesia | orang, budaya | PG |
| MS | Di Palestin umpamanya, rakyatnya yang lama dijajah dan mengalami pelbagai penderitaan dan penindasan dari segenap segi, tidak diendahkan oleh dunia antarabangsa kerana mereka dilabelkan sebagai pengganas yang mengancam keselamatan dunia | Palestin, penderitaan | MG |
| PT | Um dos primeiros costumes que você deve ter em mente na Colômbia é que o povo colombiano é um povo aberto e extrovertido, muito mais do que o povo europeu ou norte-americano | norte, extrovertido | PG |

Table 1: GeniL instance in nine languages; each sentence contains an association (pair of identity term and attribute), annotators are tasked to label each sentence either as **NG**: Not Generalizing, **PG**: Promoting a Generalization, or **MG**: Mentioning a Generalization.

in societal knowledge (Dev et al., 2023b), tend to be shallow in nature, often relying on a handful of templates (Dev et al., 2020) or co-occurrence of stereotypical associations (Bhatt et al., 2022; Jha et al., 2023). The former instantiates a limited set of linguistic manifestations of stereotypes, while the latter does not verify if all such co-occurrences are evoking the semantics of associated stereotypes in text.

One of the main ways in which stereotypes are reflected in language is through expressions of generalizations. While generalizations may sometimes be implicit (often tested through evaluation approaches such as natural language inference Dev et al. (2020)), they are quite often explicitly expressed in text, in various contexts, with varying tones and sentiments (as shown in Table 1). Furthermore, we can identify two high-level distinctions in such linguistic contexts, as per functional linguistic theory (Halliday, 1973) that posit two main purposes for language: to *express* ideas and to *influence* people. In context of generalizing language these dimensions map to: (1) is the sentence mentioning a generalization?[1], or (2) is the sentence promoting a generalization with an intention to influence others. This distinction is crucial in how they may be dealt with. For instance, model creators aiming to prevent their LMs from perpetuating stereotypes may focus only on the latter case, while those curating datasets might have to tackle both even if different strategies are applied.

In order to address this critical need, we build a large multilingual dataset of over 50K sentences from 9 languages —English, Arabic, Bengali, Spanish, French, Hindi, Indonesian, Malay, and Portuguese— annotated (by native speakers of each language) for instances of generalizations, and whether they are merely mentioning them or promoting them. Using this dataset, we demonstrate that the likelihood of a co-occurrence being an instance of generalization is usually low, and varies across different languages, identity groups, and attributes, underscoring the inadequacy of shallow approaches that rely on co-occurrence

---

[1]Note that stereotype and generalization are **not** being used interchangeably in this paper. While generalization is making broad statements about groups, stereotypes are simplified ideas about people that are known to exist in society.

metrics. Finally, to perform this task at scale, we also build classifiers that can detect generalization in language with an overall PR-AUC of 58.7, with varying degrees of performance across languages. GeniL data and classifiers can be further coupled with dynamic repositories of societal stereotypes to create more flexible stereotype detection classifiers for safety filtering and evaluation of language technologies.

## 2 Background

Stereotypes widely impact humans in their everyday lives (Quinn et al., 2007). Social psychological studies of stereotyping have provided different frameworks for explaining this process as well as its dimensions (Fiske et al., 2018; Koch et al., 2016; Abele and Wojciszke, 2014; Osgood et al., 1957). As language models learn the representation of real-world knowledge through human language, they capture and embed human biases, including stereotypes (Caliskan et al., 2017; Garg et al., 2018; Charlesworth et al., 2021). As the result, these models are highly prone to biased representations of different social identities, such as race, gender, and age, and their intersections (Basta et al., 2019; Kurita et al., 2019a; Hutchinson et al., 2020; Tan and Celis, 2019). Impact of such biased representations also extends to downstream language understanding tasks (Sheng et al., 2019b; Dev et al., 2020; Kirk et al., 2021; Davani et al., 2023).

Approaches to detect and mitigate stereotypical representations rely on various benchmark datasets. Existing benchmarks employ various methodologies for operationalizing stereotypes; in the most common approach, a stereotype is represented as an association between a pair of identity term and attribute and benchmark accordingly include various such associations (e.g., (Rudinger et al., 2018; Zhao et al., 2018a; Jha et al., 2023; Bhutani et al., 2024)). However, such benchmarks fall short on evaluating the diverse linguistic contexts in which stereotypical associations appear and instead rely on a limited set of sentence templates that fail to operationalize the nuances of biased language. An instance of flaw in conceptualization is seen when identity terms from different social categories are treated as interchangeable in stereotypical contexts, e.g., equating the gender identify term "woman" with racial/religious term "Jew". Stereotypes have also been studied on a sentence-level; the resulting benchmarks often include pairs of stereotypical and *anti*-stereotypical sentences (Nadeem et al., 2021; Nangia et al., 2020; Felkner et al., 2023). Such benchmarks fail to provide a comprehensive coverage of the various linguistic concomitants of stereotypes e.g., implicit notions, emotions, sentiments, and even behaviors (Parrish et al., 2022). As a result, such benchmarks often face criticisms for their limitations in *conceptualizing*, *covering*, and *operationalizing* stereotypes (Blodgett et al., 2021). Moreover, no single benchmark can capture the nuances of social dynamics that underlie stereotypical thoughts and behaviors over time in various cultural and linguistic contexts (Hutchison and Martin, 2015; Rauh et al., 2022). Our work contributes an important component to the larger task of detecting stereotypes that attend to the linguistic context in which stereotypical associations appear.

Motivated by and intertwined with the efforts for detecting hate speech and offensive language, researchers have also introduced and called for stereotype detection models (Zhao et al., 2018b; Bolukbasi et al., 2016; Dev and Phillips, 2019), some of which explicitly focus on sexism (Cryan et al., 2020; Chiril et al., 2021; de Vassimon Manela et al., 2021), racism (Field et al., 2021; Waseem, 2016), or other intersectional group-based derogatory language (Ma et al., 2023a; Cheng et al., 2023). Exploratory research has exposed the presence of stereotypes in various language technologies ranging from search engines (Choenni et al., 2021) to contextualized word embeddings (Tan and Celis, 2019) and more recently in large language models (Ma et al., 2023b; Jeoung et al., 2023). While existing models for detecting stereotypical language often focus on specific contexts and domains, our novel contribution lies in defining a generalizing language detection task. This task empowers us to develop detectors that can effectively identify language across various generalizing linguistic contexts. By coupling these detectors with diverse stereotype association benchmarks, we can comprehensively address the challenge of stereotype detection in language technologies. This approach not only enhances the robustness of detection methods but also fosters a more comprehensive understanding of the multifaceted nature of stereotypes in language.

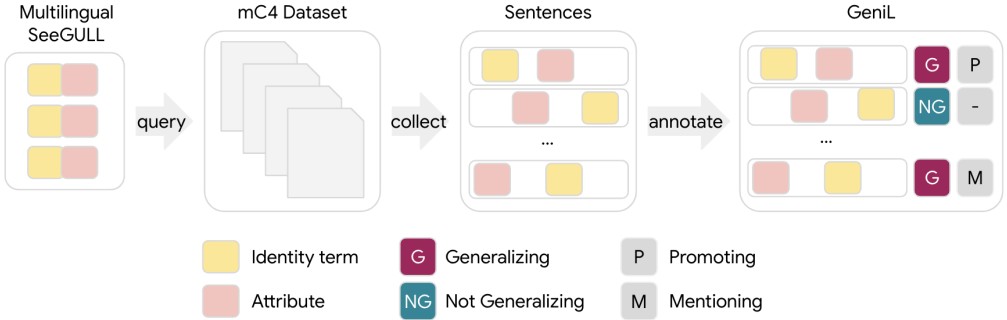

Figure 1: The process of creating GeniL dataset; we used stereotypical associations (pairs of identity term and attribute) from the Multilingual SeeGULL (Bhutani et al., 2024), and query the mC4 dataset to collect sentences which mention those pairs. During a data annotation process with trained annotators we collected two labels for each sentence: (1) whether the sentence is generalizing, and if so (2) is it promoting a generalization or mentioning.

As a socially and culturally significant question, research on detecting stereotypes in multilingual data requires focused knowledge about specific social contexts. Through community engagement efforts, Dev et al. (2023a) collect an extensive resource of stereotypes from the Indian context. B et al. (2022) extends the well-known Word Embedding Association Test to Hindi and Tamil languages by incorporating societal information about Indian social groups. Khandelwal et al. (2023) introduces a dataset of stereotype and anti-stereotype sentences in Hindi, and show that "unbiased" models still embed biases regarding caste, which is a less studied social dimension comparing to gender and age. In a multilingual effort, Bourgeade et al. (2023) curate a dataset of stereotypical language about immigrants in three languages. Steinborn et al. (2022) create a multilingual dataset for assessing gender stereotypes in English, Finnish, German, Indonesian and Thai, by translating the CrowS-Pairs dataset. Our work contributes to this line of work that expands evaluation resources to a broader set of languages, beyond English. While detecting generalizing language primarily focuses on linguistic cues, social and cultural factors also play a crucial role in stereotyping. Hence, we also collected sentences in nine languages and recruited native speakers as annotators, allowing us to examine the cross-cultural variability of this task.

## 3 The Generalization in Language (GeniL) Dataset

Our approach towards building the Generalization in Language dataset is outlined in Figure 1. For the purposes of this paper, we define *generalization* as a statement about a group of people (*identity term*), suggesting that certain characteristics or behaviors (*attribute*) apply to all members of that group. We consider generalizations in language as a means of social stereotypes in action; however, not all generalizations are existing stereotypes in society (e.g., one may say "all X are Y" without there being any evidence of X and Y being associated in society). Our focus in this paper is at the linguistic level; we are agnostic to whether a particular generalization represents an existing social stereotype nor do we assess whether it is positive or negative. However, in Section 4 we do include some analyses where we use external resources to perform some analyses along these dimensions.

### 3.1 Curating Sentences

Our first step is to curate a set of naturally occurring sentences that are likely to contain generalizations about social groups. For this we first collected a list of (*identity term*, *attribute*) tuples from the Multilingual SeeGULL (SGM) dataset (Bhutani et al., 2024), which provides thousands of such LLM generated tuples (referred to as *associations*) that are then validated by native language speakers as to whether they are known *stereotypes* in their society or not, and how offensive each attribute is. We used SGM tuples in 9 languages: English, Arabic,

| Language | Associations | Sentences | Annotators | Fleiss Kappa | %Generalizing | | |
|----------|-------------|-----------|------------|--------------|-----|-----------|---------|
| | | | | | All | Promotion | Mention |
| English (en) | 1091 | 11422 | 11 | 0.87 | 7.7 | 6.0 | 1.6 |
| Arabic (ar) | 1104 | 4803 | 7 | 0.44 | 5.1 | 2.3 | 2.8 |
| Bengali (bn) | 782 | 4941 | 8 | 0.46 | 8.7 | 5.1 | 3.6 |
| Spanish (es) | 1348 | 4965 | 5 | 0.63 | 3.5 | 2.0 | 1.5 |
| French (fr) | 1071 | 4875 | 7 | 0.56 | 8.0 | 5.7 | 2.3 |
| Hindi (hi) | 784 | 4993 | 3 | 0.71 | 1.9 | 1.3 | 0.6 |
| Indonesian (id) | 590 | 4905 | 4 | 0.48 | 2.5 | 1.7 | 0.7 |
| Malay (ms) | 634 | 4969 | 6 | 0.62 | 7.7 | 3.7 | 4.0 |
| Portuguese (pt) | 1357 | 4965 | 11 | 0.58 | 5.4 | 3.8 | 1.6 |

Table 2: Statistics of data annotated in each language

Indonesian, Spanish (Mexico), Malay, Portuguese (Brazil), Hindi, and Bengali (Bangladesh).[2] For each language, we queried the Multilingual Common Crawl (mC4) language corpus to collect naturally occurring sentences that contain both the terms in the tuples present in SGM. To ensure a diverse representation of different associations in our data, we limit the number of sentences per tuple to at most 15. The number of associations we found a match in mC4, as well as the total number of sentences for each language is shown in Table 2.

### 3.2 Annotating Generalization in Language

Our task is to detect instances where the *identity term* and *attribute* are present in any given sentence. To capture the nuances of the linguistic contexts in which generalizations manifest, we rely on the functional linguistic theory (Halliday, 1973) that posit two main purposes for language: to *express* ideas and to *influence* people. Following this, we introduce two types of generalizing language: **Promoting**: language that explicitly states and endorses a generalization (e.g., "The *Swedes* know how to stay warm and *stylish* in cold weather.") and **Mentioning**: language that references a generalization without necessarily explicitly endorsing and promoting it (e.g., "For those who think the *Irish* are a *hard-drinking* but jolly race, this play will bean eye-opener.").

Annotators were given the sentences without marking the identity term or attribute, and asked to identify if the sentence contained any generalizations (G) or not (NG) about any identity groups. If they answered that there was a generalization, they were prompted to (1) identify the identity term as well as the attribute, and (2) distinguish the generalization type as promoting (PG) vs. mentioning (MG). In other words, annotators make a high-level G vs. NG distinction, and then if they chose G, they make a further PG vs. MG distinction. In both decision points, we allowed the annotators to signal that they were unsure. Please refer to the Appendix A for a full description of the annotation manual and additional guidelines.

We assigned 3 annotators to label each sentence. Annotators for each language were native speakers, with the exception of English annotations conducted in India, with English proficiency as a selection criteria. For Spanish, Portuguese, and Arabic, we sought annotations from native speakers in Mexico, Brazil, and Saudi Arabia, respectively, since we chose source tuples from SGM corresponding to those countries. Annotators were recruited through a proprietary platform, and compensated in accordance to their local law, and were informed of the intended use of their annotations. The annotation process was closely monitored by the authors and recruiters fluent in the respective languages. We ensured task clarity by conducting initial annotation tests (with 50 items) and iteratively enhancing the annotation guide to respond to any recurring ambiguities (see "Notes" paragraph in Appendix A.1). Inter-annotator agreement rates are provided in Table 2, demonstrating moderate to substantial agreement across different languages. For the analyses presented in this paper, annotations were aggregated, and sentences were classified as "Generalizing" if a majority of annotators concurred. However, following (Prabhakaran et al., 2021) we will release individual annotations to enable future studies on subjective differences.

---

[2]Note that SGM has country specific versions of stereotype tuples for Spanish, Portuguese and Bangladesh; We chose to focus on one country each for each of these languages, for simplicity.

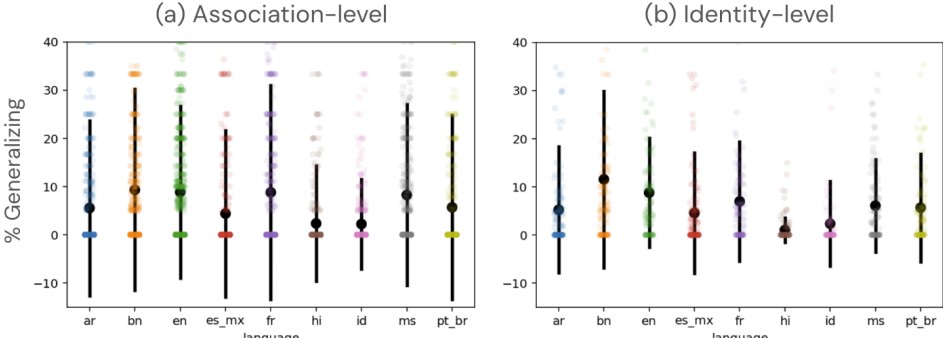

Figure 2: The frequency with which associations and identities appear in generalizing contexts varies across languages. Each color-coded point represent an association or identity and the black dot and lines respectively represent the means and standard deviations. The high deviations from the average suggests that relying solely on the co-occurrence of stereotypical associations may result in differential misclassification rates for different associations and identity terms.

## 4 Analyses

### 4.1 Generalization likelihood across associations, identities, and languages

One of the core motivations for building GeniL is that the mere co-occurrence of an identity term and attribute in a sentence does not always convey a generalization context. In fact, our dataset demonstrates that only a very small percentage of sentences with both the identity term and attribute are generalizing in nature: on average 5.9% ($SD$ = 2.4%) of our sentences across all languages are labeled as G, with the highest of 8.7% for Bengali sentences and the lowest of 1.9% for Hindi – see Table 2). In other words, any co-occurrence based approaches to estimate the extent of generalization will be over-estimating by a factor of 10 or more. Another related question is whether the rate at which each (identity term, attribute) pair occurs in generalizing context is similar — in which case one could simply scale down the estimates by a factor of 10. To answer this, we computed the mean and standard deviation for the generalization likelihood of each association as well as each identity term. We observe significant variance in generalization likelihood among different associations (Figure 2, 5). In fact, different identity terms also have huge variance in generalization likelihood suggesting that any co-occurrence based approaches are likely to incur false positives at different rates for different identity terms, which could further introduce undesirable biases. It is also important to note that there is considerable variation in generalization likelihood across different languages. Collectively, these results provide empirical support for the need for assessing stereotypes in the linguistic context of generalization for accurate evaluation.

### 4.2 Trends in stereotypical generalizations

We now delve deeper into the subset of associations that were validated to be stereotypical in respective regions as per SGM (Bhutani et al., 2024). Here, we treat any association that was labeled as an existing stereotype by at least 2 out of 3 annotators in SGM to be a stereotypical association. The generalization likelihood is higher for associations that are identified as stereotypical in SGM (6.5%) compared to those that are not (4.8%). Similar to the general trend, stereotypical associations are also more likely to appear in non-generalizing language compared to generalizing contexts, across all languages (see Figure 3(a)). The varying distributions observed in Figure 3 can be interpreted as potential evidence of the various manifestation of stereotypes and generalizations in different linguistic and cultural contexts. It is possible that generalizations are more prevalent, explicit, or easily detectable in French, Bengali, or Malay sources compared to others. Alternatively, the observed variations might reflect variations in the representation of each language in mC4, including factors such as sample size, or source selection. We further explore the ratio of offensive attributes

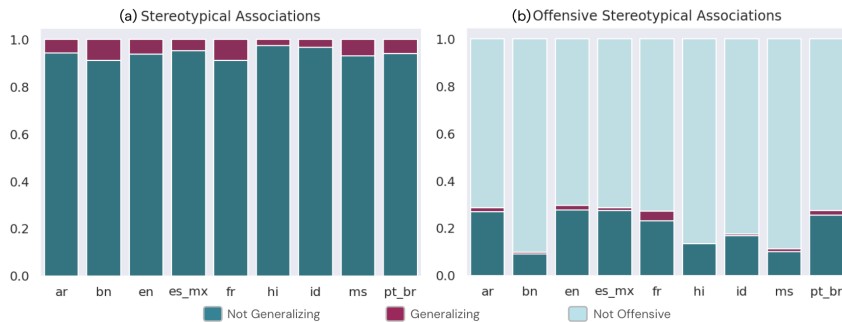

Figure 3: Emergence of associations in generalizing and not generalizing language. (a) shows the likelihood of stereotypical associations appearing in different contexts, in almost all languages this ratio in smaller than 10%. (b) focuses on stereotypical associations that are marked as offensive. Offensive stereotypical associations are extremely unlikely to appear in generalizing language.

being used in generalizing contexts across different languages. Figure 3(b) shows the distribution of offensive attributes across generalizing sentences. Across different languages, when sentences mention an offensive stereotypical association they are on average 7.8% (SD = 3.9%) likely to be generalizing (even higher rate than just all stereotypes). These findings substantiates our claim that filtering or evaluating text based on co-occurrences of stereotypical association is a biased approach, and attending to the linguistic context, in which a stereotypical association appears, has vital importance for detecting stereotypical language and downstream harms.

### 4.3 Promoting vs. Mentioning distinction

As outlined earlier, even when associations appear in generalizing language, there might not be an explicit intention for promoting an stereotype. Some such contexts include associations that appear in sentences meant to inform, or negation of stereotypes. We found that the ratio of promotion to mentions vary across different languages. In general, generalizing language is more likely to be promoting (Mean = 62.87%, SD = 11.40%) than mentioning a generalization, this likelihood is highest in English (78.72%), while in Arabic (45.12%) and Malay (48.31%), the mentioning is slightly more probable. This distinction is especially important if the model creator intends to mitigate stereotyping by filtering stereotypes from training set vs. filtering outputs through safeguards; in the latter case the mentioning cases are arguably okay to be not filtered out.

## 5 Multilingual Classifiers for Generalizing Language

Our analyses emphasized the crucial role of the linguistic context for detecting stereotypical generalizations in text. Now we turn to how well the GeniL dataset enables us to automatically detect generalizing language. Our primary objective here is to establish a reasonable baseline performance on this task, while also demonstrating the necessity and importance of collecting training data in diverse languages through our efforts to foster comprehensive generalizing detection in different languages.

We trained classifiers by fine-tuning two well-known multilingual large language models, mT5-XXL and PaLM-2-S. The train/val/test split for each language consist of 70%, 10%, and 20% of the data, respectively. We used a batch size of 32 and trained mT5 for 10000 steps and PaLM-2-S for 5000 steps with learning rate 1e-3 and 1e-4 respectively with dropout 0.05 (based on prior hyper parameter experiments using these models). In order to assess the utility of multilingual annotations in GeniL, we conduct experiments assessing the performance of classifiers on all 9 languages under three training set configurations:

| Model | Training set | all | en | ar | bn | es | fr | hi | id | ms | pt |
|---|---|---|---|---|---|---|---|---|---|---|---|
| mT5 | **en** | 45.1 | **70.8** | 19.1 | 7.9 | 66.1 | 65.2 | 54.8 | 24.4 | 27.0 | 54.5 |
| | **en+translated** | 45.8 | 66.1 | 23.9 | 8.6 | 55.9 | 60.6 | 56.7 | 19.4 | 36.1 | 58.4 |
| | **multilingual** | **57.8** | 67.3 | **48.2** | **15.6** | **71.9** | **73.0** | **65.7** | **31.7** | **58.9** | **65.3** |
| PaLM | **en** | 46.4 | **67.3** | 30.1 | 13.7 | 67.6 | 60.9 | 52.8 | 20.0 | 34.2 | 49.5 |
| | **en+translated** | 42.6 | 58.9 | 21.6 | 7.8 | **70.4** | 60.0 | 44.4 | 29.9 | 37.7 | 52.6 |
| | **multilingual** | **57.9** | 67.3 | **46.5** | **17.8** | 68.5 | **75.6** | **64.9** | **43.2** | **57.9** | **63.0** |

Table 3: The results of multilingual generalizing language detectors. We trained two base models mT5(-XXL), and PaLM(-2 S) through three training configurations. All models are then tested on different languages and the performance (calculated as the PR-AUC) shows that using the multilingual GeniL leads to best performance in almost all languages.

- **en**: training only on English data from GeniL. This setting emulates the case where there are limited resources and we only have the English GeniL data and want to assess how much performance degradation happens.

- **en+translated**: augmenting English data with (machine[3]) translations into the target language. This setting also emulates the case where there is only English GeniL data, but mitigating the multilingual gap through automated translations.

- **multilingual**: training on annotated data in the target language. This setting demonstrate the full use of GeniL data.

Table 3 presents the PR-AUC values obtained for all 9 languages in all three settings. First of all, the best performance obtained on both base models varies substantially across languages. There was no substantial upper hand for either base model. While classifiers in English, French, and Spanish posted relatively high PR-AUC values, those in Bengali, Arabic and Indonesian posted the lowest PR-AUC values. It is interesting to note that this trend is in line with the inter-rater agreement — Bengali, Arabic and Indonesian data also obtained the lowest annotator agreement scores. This could mean that either the data is of lower quality for these languages, or that the task itself is harder in these languages for both humans and classifiers. Future work should look into targeted efforts to improve these results.

Next, we look into how our different training configurations fared. First, we observe that using English GeniL data alone (i.e., **en**) results in substantial drops in PR-AUC values on all languages (sometimes even 30+ points) except for English (which is expected). While the **en+translated** setting bridged this gap in some cases, it caused regressions in several languages compared to training on **en** annotations alone. Training on annotated data in multiple languages (**multilingual**) remedies these issues and obtains the best performance across all languages (without much regression for English). These classification results clearly demonstrates that in order to effectively detect generalizing language in multiple languages, annotated data in those languages is essential.

Finally, we performed the same set of experiments using the base models and training configurations for three additional tasks: (1) predicting whether the generalizing language is Promoting vs. Mentioning an association, (2) determining which is the identity term, and (3) what is the associated attribute (Figure 4). In other words, these experiments evaluate performance on various sub-tasks in an end-to-end setting, where the input sentences do not have either the identity term or attribute marked. We observe a similar trend in performances as above, where Bengali, Arabic, and Indonesian continue to post the lowest scores, while English, French, Spanish and Portuguese posted relatively high performance on making the PG vs. MG distinction in all three training settings. However, when it comes to detecting the identity terms and attributes, the performance drops significantly for French, Spanish, and Portuguese, although the multilingual setting bridges the gap in all three of those languages substantially. In fact, the gains from multilingual data is substantially larger in the tasks of identifying attributes and identities for all languages.

---

[3]using the publicly available Google Translate API

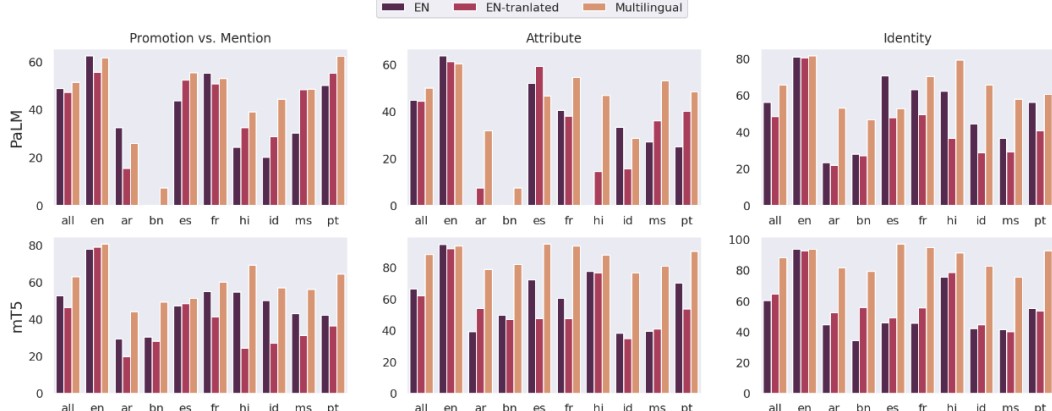

Figure 4: The results of multilingual classifiers on three tasks: (1) is the generalizing language Promoting or Mentioning an association, and what are the (2) identity term, and (3) attribute that are shaping the association. Models are tested on different languages and the performance (calculated as the F1-score) shows that using the multilingual GeniL leads to best performance in almost all tasks and languages.

## 6 Discussion and Conclusion

We introduced the new task of detecting generalization in language to enable nuanced stereotype evaluations of language technologies. We also presented GeniL, a multilingual dataset of over 50K sentences (English, Arabic, Bengali, Spanish, French, Hindi, Indonesian, Malay, Portuguese) annotated for instances of generalization. Our analysis reveals that co-occurrence metrics are unreliable proxies of generalization, and we built automatic classifiers to detect generalization with reasonable performance (overall PR-AUC of 58.7). We anticipate the GeniL tagger(s) enhancing different aspects of the ML pipeline:

- *Nuanced Evaluations*: In the context of assessing the extent to which certain generative language models perpetuate stereotypes, the GeniL tagger provides a nuanced view that takes into account the sentential context, essentially enabling the evaluations to focus on instances where the model promotes potentially problematic generalizations rather than mere co-occurrences.

- *Targeted Safeguards*: If a model creator intends to put safeguards to prevent generative models from stereotyping language, currently the only brute force approach is to remove any generations that evoke certain terms that are known to be stereotypically associated with identity terms mentioned in text. The GeniL taggers enable a more nuanced approach which can more precisely identify problematic generations that promotes stereotypes.

- *Enhanced Data Curation*: If a model creator intends to intervene by filtering out or balancing the presence of stereotypical associations in the training data, then GeniL enables a more fine-grained approach that focuses on data instances that has generalizing language. GeniL tagger will also enable the inclusion of such targeted statistics in data transparency artefacts (e.g., the SoUND Framework (Díaz et al., 2023)).

### 6.1 Limitations

In this paper, we focus on cases of explicit generalizations that happen in the same sentence. However, generalization may sometimes happen over multiple sentences. For instances the two sentences "I met a French man at the train station. Of course, he was rude!" are together perpetuating the stereotype that French people are rude, however this does not happen in the same sentence. GeniL will miss such instances, despite them being made explicit in language. Future work should expand into such discourse level generalizations. We also do not capture implicit generalizations that do not have both the identity term and the attribute mentioned in text.

GeniL is not intended to be a full representation of all potentially stereotypical or generalizing sentences in the target languages. It is a focused lens on the diverse linguistic contexts of generalization. While SGM, a large validated dataset of stereotypes, is an excellent starting point, we do recognize its inherent limitations and biases.

Moreover, whether or not a sentence has generalizing language has some level of subjectivity. While we ensured gender balance in our rater pool, it may have skews along other socio-demographic axes. We used majority vote in this work to arrive at a ground truth label, however future work should look into individual and group level differences. We release individual annotations to enable any such follow up studies.

While the main contribution of this paper is the task and the dataset, our classifier experiments suggest that the task of detecting generalization as well as distinguishing the two kinds are both non-trivial tasks computationally. The overall PR-AUC is 58.7 which suggest ample opportunities for improvement. In particular, the performance is the poorest for Bengali, followed by Indonesian and Arabic (all three posted PR-AUC less than 50), which are also the languages that obtained the lowest inter-rater agreement. It can be interpreted that this task is more challenging for humans to annotate in certain languages, due to factors such as linguistic nuances or cultural differences. This in turn could make it more difficult for the model effectively in those languages. Future work should look into improving both the data quality and model performance in these languages, as well as expand coverage to other low resource languages. Future work should also expand GeniL to other languages with diverse morpho-syntactic features to ensure more robust coverage across language families.

## Ethics Statement

We build GeniL dataset and classifiers with the intention to be used for evaluation purposes. By design, the dataset contains a high incidence of potentially problematic generalizations about groups of people. While this corpus enables large-scale and finer-grained stereotype evaluation, the concentration of such sentences with potentially harmful associations may be triggering for certain individuals. Hence care must (and will) be taken in how the data is disseminated and used downstream.

There is a risk of misuse wherein the GeniL tagger not detecting an instance of generalization is deemed as an absence of generalization. In addition to the error in recall, it is also important to remember that GeniL focuses only on explicit generalizations, and instances of implicit generalization may still be present. Finally, there is a potential for malicious use where someone could use the GeniL dataset to intentionally generate generalizing language targeting specific identity groups.

## Acknowledgements

We thank Akshita Jha for early discussions on this project. We also appreciate Kevin Robinson and Kathy Meier-Hellstern for useful feedback on early versions of this manuscript. We thank the annotation team for facilitating our data work.

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

# A  Appendix

## A.1  Annotation Manual

**Objective**   This document provides guidelines for Generalizing Language Annotation. We define generalization as a broad statement made about a group of people, suggesting that certain characteristics or behaviors apply to all members of that group.

**Labels**

- Generalizing (G): a broad statement is being made about a group of people. Please note this statement can be positive, negative, or neutral.
    - Promoting (PG): Language that explicitly states and endorses a stereotype or generalization about a group of people. (e.g., "The Swedes know how to stay warm and stylish in cold weather.")
    - Mentioning (MG): Language that references a stereotype without necessarily explicitly endorsing and promoting it. (e.g., "She says Namibians, in general, are not lazy, but they rely too much on the government to do everything for them.")
    - Other (OG): Language is generalizing but it is not clearly an evocation or mention, can be other kinds of generalization.
- Not Generalizing (NG): Language that does not mention a group of people, or mentions a group of people without evoking or mentioning a generalization
- Can't Decide (C): Language that is vague, ambiguous, and unknown, such that you could not decide whether there is a generalization (please provide justification).

If Generalizing (G) is selected:

- Who is the generalization about? Highlight the part of text that mentions the group
- What is being generalized? Highlight the attribute (preferably one word) that is used to describe this group.

**Note:**

- Please read the text fully.
- All provided texts include at least one word that can be an identity term (e.g., "French", "Iranian").
- Decide whether the text is about a general assumption about a group.
- The generalization can take different forms, it can be positive or negative, moral or immoral, accurate or inaccurate.
- If the language is explicitly making a general statement or stereotypes a group (e.g., example 2), label it as Promoting (PG).
- A sentence can still be Promoting (PG) even if the generalization is not the main focus of the sentence (e.g., example 1).
- If the text mentions a generalization without necessarily endorsing it (e.g., "although everyone thinks French people love cheese, he hates cheese"), label it as Mentioning (MG). Look for language about commonly held beliefs or assumptions about a group, even if it doesn't directly state them as facts.
- To decide between Promoting (PG) and Mentioning (MG), ask whether the sentence suggests that the author believes the generalization? If so, the sentence is Promoting (PG), but if the generalization is not what the author is necessarily agreeing with (e.g., it's a quotation, it's reporting something said or believed by other people etc.) it should be labeled as Mentioning (MG).
- In exceptional cases where you can't decide between Promoting (PG) and Mentioning (MG) you can select Other (OG).

| | Sentence | Question: Does the text make a generalization? Explanation | Label |
|---|---|---|---|
| 1 | Poverty spreads from cities to countryside despite the people of Laos' hardworking perseverance. | The sentence states that all people of Laos are hardworking. | PG |
| 2 | It turns out the French aren't just good at making wine, but they can make a mean cider as well! | The sentence argues for a generalization (French are good at making wine and cider) | PG |
| 3 | For those who think the Irish are a hard-drinking but jolly race, this play will be an eye-opener. | The sentence suggests that a generalization exists (Irish are hardrinking and jolly), but does not promote it. | MG |
| 4 | We all enjoyed an amazing French Day last week with lots of singing as we learnt colours and numbers in French and ate cheese and bread in our lovely blue, white and red outfits! | | NG |
| 5 | Travelers will find Turks to be exceptionally gracious hosts. | The sentence implies that Turks are gracious hosts. | PG |
| 6 | Representative Maritza Davila said that the Turks are the most diligent among the ethnic communities in New York. | The sentence mentions a generalization made by a person, and does not promote it. | MG |
| 7 | If you want to add some new moves to your wrestling arsenal then be sure to pay a visit to Randall Lovikiv, the hard drinking Russian. | The sentence mentions a hard drinking Russian, does not generalize any attribute to all Russians. | NG |

Table 4: Examples provided to the annotators

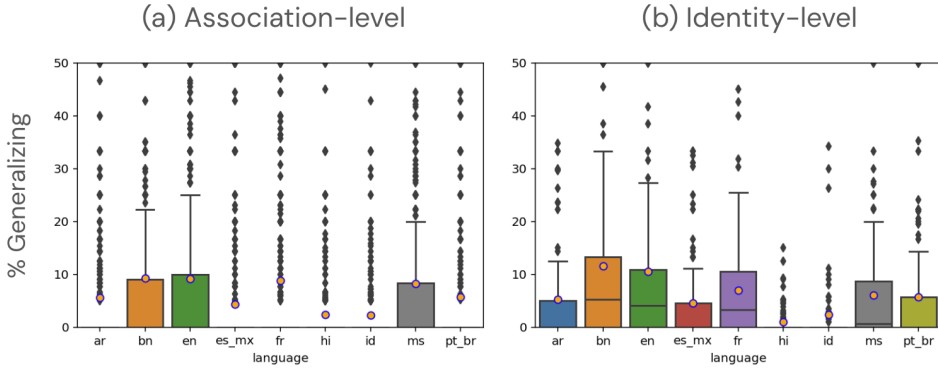

Figure 5: The frequency with which associations and identities appear in generalizing contexts varies across languages. This is the same info as Figure 2 using a box plot

## A.2 Distribution of Generalizing language

Table 5 shows the number of GeniL sentences across different languages that mention a stereotypical or non-stereotypical association (SGM), separated by their generalizing (G) or non-generalizing (NG) context, determined in our annotations.

| | Stereotypical Association | | Non-stereotypical Association | |
|---|---|---|---|---|
| | Generalizing | Not Generalizing | Generalizing | Not Generalizing |
| ar | 208 | 3511 | 38 | 1046 |
| bn | 359 | 3710 | 71 | 801 |
| en | 474 | 5572 | 382 | 4592 |
| es | 109 | 2189 | 66 | 2652 |
| fr | 237 | 2572 | 154 | 1912 |
| hi | 50 | 1951 | 45 | 2947 |
| id | 66 | 2008 | 55 | 2776 |
| ms | 277 | 3864 | 115 | 983 |
| pt | 261 | 4108 | 5 | 604 |

Table 5: Distribution of sentences mentioning various associations across generalizing and non-generalizing labels.

