# OpenReview forum: "GeniL: A Multilingual Dataset on Generalizing Language"
_colmweb.org/COLM/2024/Conference — COLM_

### Official Review · Reviewer_wMqL · 2024-05-08

**Rating:** 8
**Confidence:** 4
**Ethics Flag:** 1

**Summary:**

This submission introduces a corpus for testing (stereotype) generalization in multiple languages and builds classifers to detect generalizing language, demonstrating clear improvement over template- and cooccurrence-based approaches.

**Questions To Authors:**

Assuming guidelines were applied similarly in each language, how do you explain the large disparity between languages in the analysis and results (e.g. Fig. 3 or Tab. 3).

I would recommend using box-and-whiskers plots in Figure 2 rather than mean and standard deviation, which are not that informative for strongly skewed distributions.

I was confused by "respective societies" in Sec. 4.2 -- can you clarify or rephrase? Also later in that Section: "that that", "a stereotypical associations"

[Read reviews and rebuttal -- thanks]

**Reasons To Accept:**

1. A clear, well-motivated and well-grounded definition of the generalizing language task, covering different usage.
2. A novel dataset operationalizing this task definition, using native speakers as annotators (not LLMs!), across 9 languages, with interesting insight along the way, e.g. regarding how prevalent generalizing language is. The dataset will be released with multiple individual annotation to encourage further analysis.
3. Multilingual generalizing language detectors, providing a reasonably high performance baseline on most languages for this task.

**Reasons To Reject:**

There is clearly some issue with either the data or the annotation in some languages that cause a large disparity in performance and would warrant further analysis. The last paragraph of Section 5 would deserve more detail and clarity. Apart from that, this is a clear paper with a well defined goal, a valuable contribution, and well-done experiments providing interesting and non-trivial insight; I see really no reason to reject.

---

> ### Author Rebuttal · Authors · 2024-05-31
>
> We genuinely appreciate your feedback and comments!
>
> We agree with your point about extending the discussion of results in Section 5 and will provide more discussion for different languages.
>
> - We interpret the variation in the results observed in Figure 3 as potential evidence of how stereotypes and generalizations manifest differently in various linguistic and cultural contexts. It is possible that generalizations are more prevalent, explicit, or easily detectable in French, Bengali, or Malay sources compared to others. Alternatively, the observed variations might reflect variations in the representation of each language in mC4, including factors such as sample size, or source selection. We will expand on our discussion and limitations section to highlight this.
> - We appreciate your  suggestion to use a boxplot. We did try boxplots, however, because of the high frequency of 0 values (social groups not appearing in any generalizing sentences), boxplots could not effectively illustrate the differences between languages.
> - Thanks for the comments, we will change “societies” to regions, and will fix the typo.

---

> > ### Comment · Reviewer_wMqL · 2024-05-31
> > **Thanks**
> >
> > Thank you for the replies -- looking forward to reading the final version.  I am slightly puzzled by your comment on the boxplots: In that case, the box should go from zero to the 3rd quartile, with whisker extending above, but not below, which would arguably give a more accurate idea of the distribution than error bars extending largely below zero.

---

> > > ### Author Response · Authors · 2024-05-31
> > >
> > > We agree with your point. The issues we meant to point out are:
> > >
> > > - The minimum, Q1, and median are 0 for most languages, but we do agree that the Q3 and whiskers are still informative.
> > > - Compared to the boxplot, our current figure shows the SD and, accordingly, variance across different languages, which was the main point of the figure. However, we do see your point that showing the boxplot could also be informative. We will add that to the appendix.
> > >
> > > We appreciate your feedback and suggestions.

---

### Official Review · Reviewer_uFUJ · 2024-05-10

**Rating:** 9
**Confidence:** 4
**Ethics Flag:** 1

**Summary:**

In this paper, the authors present a new dataset and task for "Generalizing Language," dubbed GeniL. GeniL consists of 50k sentences annotated for _generalization_. For each sentence, generalization is determined given an association pair of (_identity_, and _attribute_). The association pairs were extracted from "Multilingual SeeGULL" and the sentences containing those pairs were queried from multilingual Common Crawl. Human annotators were then asked to tag each sentence for generalization using three tags: **P**romoting **G**eneralization (PG), **M**entioning **G**eneralization (MG), and **N**ot **G**eneralizing (NG). Both PG and MG indicate generalization. GeniL is a multilingual dataset spanning **nine** languages from different language families.

While the primary contribution of this paper is the dataset, the authors also introduced a generalization classification task and benchmarked it. The main task is to classify generalization regardless of its type (PG or MG).

The major finding of this paper is the importance of the sentential context of the association pair in determining generalizations, let alone the detection of stereotypes. The authors demonstrated this by contrasting the annotator labels with previously existing stereotype labels. The authors showed that the existence of the association in the sentence rarely entails a generalization, and hence, current stereotype classification models may inflate such classes by introducing false positives.

**Questions To Authors:**

- It may be computationally consuming, but I believe part of a more comprehensive benchmark is to fine-tune the models for every language separately and evaluate them the same way as the English one was evaluated.
- Have the authors looked at the syntactic structures of the datasets? There could be useful insights regarding the correlation between certain structures and generalizations. The same goes for the correlation with the [committed] belief of the author and whether it indicates the generalization to be PG or MG.

**Reasons To Accept:**

- The paper is very well written and easy to follow.
- Contributions and novelty can seen clearly from the text.
- The different views of the data showcase its value.
- The benchmarking methodologies seem to be sound and convincing.

**Reasons To Reject:**

There are no major weaknesses or reject-warranting reasons. However, I list some points that may be concerning.

- Some of the discrepancies in the inter-annotator agreement for a language like Arabic may be directly correlated with the diversity of the cultures and regions it is spoken in, similar to English in that sense. It would be useful also to include the primary place of residence/childhood home for the annotators and try to balance annotators based on that.
- The translation model used to translate the data isn't mentioned in the paper.
- In the discussion, the authors said " ... task itself is harder in these languages for both humans and classifiers." How can we know it is hard for the classifiers if the evaluation data has already low IAA? In other words, we can deduce that it is hard for the classifiers if our evaluation data is consistent. (Maybe this is what the authors meant)

---

> ### Author Rebuttal · Authors · 2024-05-31
>
> We deeply appreciate your encouraging and helpful feedback.
>
> - Variation in place of residency: For each language we selected annotators from the same country associated with that language. However we agree that it is important to not limit a language to a specific region, especially for languages that are spoken across multiple geo-cultural regions. We will add this point to the limitations.
> - Translation: we used the publicly available Google Translation API for this task. We will incorporate the details of the translation process into the paper.
> - Task difficulty: You are right, we cannot directly conclude that the task is inherently difficult for a model. Our intention was to highlight that this task might be more challenging for humans to annotate in certain languages, due to factors such as linguistic nuances or cultural differences. This in turn could make it more difficult for the model effectively in those languages. We will clarify this.
> - Other questions:
>     - Experiment with other monolingual models: experimenting with diverse monolingual models is intriguing. The focus of our experiments were on the most prevalent approach when training data is sparse in a target language, which is to leverage English data.
>     - Syntactic structure of generalizing: The linguistic construction of generalizing language presents a fascinating avenue for future research. We acknowledge the potential insights that could be gained by exploring the syntactic patterns and structures characteristic of generalizations. We will include this as a promising direction for future investigation in our revised work.

---

> > ### Comment · Reviewer_uFUJ · 2024-06-04
> >
> > Thank you for your response.

---

### Official Review · Reviewer_meod · 2024-05-13

**Rating:** 6
**Confidence:** 3
**Ethics Flag:** 1

**Summary:**

This paper focuses on the challenge of detecting stereotypes in LLM-generated language by studying the expression of generalizations. It proposes a multilingual dataset GeniL that is annotated for generalizations. The experimental results show that the mere co-occurrence of an identity term and attribute being a generalization is generally unlikely and varies by language, identity group, and attribute. The authors also train classifiers on the proposed dataset to improve the performance of generalization detection.

**Questions To Authors:**

(1) On Page 6, how is the factor of "10" derived in "any co-occurrence based approaches to estimate the extent of generalization will be over-estimating by a factor of 10 or more."?

(2) It is interesting in Figure 3 that stereotypical associations are much more frequent in non-generalizing contexts. What could be the possible reasons?

**Reasons To Accept:**

This paper proposes a multilingual dataset for generalization, which may be useful for future research. The experiments and some analyses are interesting, although further experiments are necessary to support the findings.

**Reasons To Reject:**

(1) The technological impact of this paper is limited.

(2) I have some concerns about the GeniL dataset. The annotations are subjective. For example, the labels of the examples presented in the appendix are controversial, and the annotator agreement is weak for some languages. These inconsistency could undermine the reliability of analyses in the paper. It would be better if more experiments could be conducted to support the claims.

(3) Relevant to the previous point, the purpose of the GeniL dataset may appear unclear, given that the same association pairs occur in various contexts. Why not analyze and label the generalizations within the existing SGM dataset? If extending to a broader corpus such as mC4, why not extract pairs of identity and attributes from scratch, instead of querying sentences by using the existing (potentially biased) pairs from SGM?

(4) The method to create GeniL dataset using the existing stereotype dataset might introduce bias into some conclusions. It would be better to see some discussions on this aspect.

(5) The authors claim to propose the new task of detecting generalization to help identify stereotypes. However, there is limited evidence showing that detecting generalization does help.

---

> ### Author Rebuttal · Authors · 2024-05-31
>
> We appreciate your feedback. Some concerns may stem from misunderstandings about our objectives and data. We will clarify them in the paper:
>
> 1- We respectfully disagree with the limited technological impact. Our work addresses a critical gap in evaluations of stereotypes by introducing a novel task, a multilingual dataset (9 languages), and trained classifiers. While we use off-the-shelf classification methods, our novelty and impact (recognized by other reviewers) stem from the new task and dataset, which is within COLM's scope.
>
> 2-  We acknowledge the inherent subjectivity of the task and have addressed it in limitations.
> To mitigate this, we emphasized the identifiable linguistic patterns of generalizations, and iteratively communicated with annotators to ensure consistency, resulting in moderate to substantial agreements, supporting dataset reliability. Despite limitations, our work offers a valuable foundation for future research.
>
> 3- There seems to be a misunderstanding about the SGM dataset. SGM gathers stereotypical word pairs (e.g., "French" and "rude") without any sentential context. GeniL evaluates SGM pairs in linguistic context to determine if a sentence intends to generalize; e.g., in sentences mentioning “French” and “rude”, GeniL asks if the sentence implies French people are rude? This is crucial for assessing harmful generalizations in text.
>
> 4- GeniL is not intended to be a full representation of all potentially stereotypical or generalizing sentences. It's a focused lens on the diverse linguistic contexts of generalization. While SGM, a large validated dataset of stereotypes, is an excellent starting point, we do recognize its inherent limitations and biases, which will be discussed in our limitations.
>
> 5- We show that detecting harmful generalizations requires more than identifying stereotypical associations, because of:
> - their frequent appearance in non-generalizing language (section4.2)
> - nuances in the intended meaning and harm of generalizations (section4.3).
> - linguistic nuances varying across languages (section4.1).
>
> Questions:
> - 10x is a conservative estimate; at most 8.7% of sentences with stereotypes are labeled as generalizing. SGM-based classifiers would flag 100 / 8.7 ≈ 11.5x more sentences as stereotypes, even higher for languages with less generalization. 10x is a rounded value for simplicity.
> - Indeed figure3 shows a key observation: the surprisingly high ratio of non-generalizing contexts of stereotypical SGM pairs.

---

> > ### Author Response · Authors · 2024-06-06
> >
> > Dear reviewer, as the end of the discussion period is approaching, we wanted to reach out and ask if there's any additional clarification we can provide to address your concerns.
> >
> > As we outlined in our response, many of the concerns raised in this review stem from a misunderstanding of the dataset from prior work (i.e., SeeGULL) that we used. In particular, SeeGULL contains only tuples of stereotypical associations with no sentential contexts, while our work focuses on sentences where both terms in the tuple occurs and assessing whether the language evokes/mentions a generalization that reflects that stereotype.
> >
> > We will update the text to avoid such a misunderstanding, and welcome any suggestions towards that. If we have addressed your concerns about the paper, we kindly request you to consider updating the score to reflect that.

---

> > ### Comment · Reviewer_meod · 2024-06-07
> > **Thanks for your response!**
> >
> > Thanks for the response and clarification! It clears up several of my concerns, so I have increased my scores. However, I still have some reservations about the consistency of the experiments and the collected dataset.

---

### Decision · Program_Chairs · 2024-07-10

**Decision:**

Accept

**Comment:**

This paper proposes a multilingual dataset of sentences annotated for generalization and reports the results of a set of experiments on detecting generalizing language.

The reviewers appreciated the novel focus on generalization, the careful construction of the dataset and its multilingual character. The paper is clearly written and the approach well motivated. While some aspects could be improved (the reviews provide a few pointers), in my view the paper is overall solid and makes a valuable contribution.